# The Typology and Topography of Child Abuse and Neglect: The Experience of a Tertiary Children’s Centre

**DOI:** 10.3390/ijerph19138213

**Published:** 2022-07-05

**Authors:** Geoff Debelle, Nikolaos Efstathiou, Rafiyah Khan, Annette Williamson, Manjit Summan, Julie Taylor

**Affiliations:** 1Birmingham & Solihull NHS Clinical Commissioning Group, Birmingham B4 6AR, UK; 2School of Nursing, University of Birmingham, Birmingham B15 2TT, UK; n.efstathiou@bham.ac.uk (N.E.); rxk869@student.bham.ac.uk (R.K.); 3Birmingham Women and Children’s Hospital Foundation NHS Trust, Birmingham B15 2TG, UK; annette.williamson2@nhs.net (A.W.); manjit.summan@nhs.net (M.S.)

**Keywords:** child maltreatment, administrative data, domestic violence/abuse, physical abuse, burns, neglect, emotional abuse, poverty

## Abstract

Effective child protection systems and processes require reliable and accurate data. The aim of this study was to determine what data could be extracted from hospital records in a single site that reflected a child’s journey from admission with suspected abuse to the decisions regarding substantiation made by the multidisciplinary child protection team. A retrospective study of the case records of 452 children referred to a major UK children’s tertiary centre for suspected child maltreatment was undertaken. Child maltreatment was substantiated in 65% of referred cases, with the majority of referrals coming from children living in the most deprived neighbourhoods in the country. Domestic violence and abuse and the child’s previous involvement with statutory bodies were associated with case substantiation. Physical abuse predominated, with soft tissue injuries, including dog bites and burns, most frequent. Burns were related almost exclusively to supervisory neglect. There were also cases of medical neglect. Emotional abuse was associated with exposure to domestic violence and abuse and to self-harm. The strengths and limitations for single-centre data systems were explored, concluding with a recommendation to establish an agreed national and international minimum data set to protect children from maltreatment.

## 1. Introduction

Child maltreatment requires a child rights and public health response [1]. This is reflected in policy and the typologies of responses in various jurisdictions within and across countries [2]. Robust data systems are required to support such responses, yet without a common agreed definition of what constitutes child maltreatment, and an accompanying minimum data set, comparisons between different typologies, while desirable, is not possible. 

Different child protection system typologies reflect an individual country’s underlying culture and professional structures of its jurisdictions. These in turn reflect how they conceptualise children and families and are therefore in a state of evolution. However, all typologies embrace early identification and prevention and share a legal framework for judging harm; it is only the nature of the response that vary, from formal, statutory legal requirements to legal protection using informal solutions within the context of community action and engagement [2]. In many jurisdictions, such as the UK, USA and Australia, professionals have a duty of care to act on ‘reasonable suspicion’ that a child has been harmed or is at risk of being harmed, and in some jurisdictions, this duty is mandated by law. Then, a child protection investigation takes place to decide whether suspicion has been substantiated, and if so, the legal and supportive framework under which the child will be protected from harm. In England and Wales, abuse is substantiated when there are reasonable grounds to suspect that the child may continue to suffer, or is likely to suffer, ‘significant harm’, i.e., ill treatment or impairment of health and development by a carer as compared with the care that would reasonably be expected of a similar child [3]. If this legal test is met, the child is placed on a child protection plan.

The World Health Organization (WHO) and International Society for Prevention of Child Abuse and Neglect (ISPCAN) call for a common conceptual and operational definition of child maltreatment [4]. They define it conceptually as ‘all forms of physical and/or emotional ill-treatment, sexual abuse, neglect or negligent treatment or commercial or other exploitation, resulting in actual or potential harm to the child’s health, survival, development or dignity in the context of a relationship of responsibility, trust or power’, and state that this ‘must be translated into operational definitions using a universally accepted classification system, such as the International Classification of Diseases (ICD)’.

ICD is the standard system for epidemiology research and health management purposes [5]. Many countries publish data on reported child maltreatment based on the hospital record at the time of the child’s discharge. However, studies across many countries and jurisdictions have shown that the use of ICD has been associated with under-representation of cases compared with other data sources, such as hospital records or linked data from other agencies [6,7,8]. One of the many reasons given for this discrepancy has been clinician uncertainty, with the outcome not becoming apparent until investigations have been complete, weeks or months after discharge. This has lessened with the addition of codes for child maltreatment-related features, ‘possible abuse’ or ‘suspected abuse’ or ‘case under review’ with periodic revisions of ICD, and with the use of data linkage [9,10,11]. 

In England and Wales, official statistics reflect children on a child protection plan; at the time of our study, the rate was 3.78/1000 live births [12]. Other jurisdictions also publish rates for substantiated abuse from local agencies. Within hospital settings, decisions around case substantiation are made within multidisciplinary child protection teams, based on assessment of risk to the child. While there is general consensus around risk of harm to the child within the legal framework of the jurisdiction, decision making may be subject to variation in thresholds of risk, bias and constraints existing within the team [13,14]. While analysis of decision making by the hospital team was considered beyond the scope of this study, the distinction between substantiated and unsubstantiated cases was retained as a potential benchmark for comparison with teams within and between jurisdictions.

### 1.1. Background

Gonzalez-Izquierdo et al. [7] asked whether ICD data can reliably or accurately reflect clinical concerns or actions by health professionals, data that could only be gained by data extraction from the clinical record, with linkage to social care data on substantiated cases. This applies particularly to neglect, the commonest form of substantiated maltreatment in England and Wales [12], where it might be subsumed under an ICD code for the injury (e.g., burn) for which the child was admitted, without due consideration to factors in causation such as supervisory neglect. Moreover, there are no specific ICD codes for the different typologies of neglect [8,11]. The methodology proposed by Gonzalez-Izquierdo et al. [7] was used in this retrospective study of children and young people admitted to Birmingham Children’s Hospital, a large tertiary children’s centre in the UK from 2011–2014.

Birmingham is the largest city in the UK outside of London. It is rich in cultural diversity and has a relatively young population, many of whom live in poverty. Birmingham Children’s Hospital serves a local area and provides tertiary services regionally and nationally. At the time of the study, the child population of Birmingham was 259,000 [15]. The rates of children with substantiated abuse in Birmingham and across the UK remained relatively constant during the first 2 years of the study period (2011–2012) but increased by 13.5% in 2013–2014 [12]. 

Around 6% of children 16 years or younger in Birmingham will have an unplanned admission to the hospital in any one year. This provides a unique opportunity to determine whether routinely collected data on a child’s journey through the child protection system and related socioeconomic and contextual factors can be used to develop a standardised minimum dataset, with agreed codes that encompass both typology and topography of abuse, to enable comparisons between the UK and other countries. 

### 1.2. Aims

The aim of the study was to determine what data could be extracted from hospital records that reflects the child’s journey from admission with suspected child abuse to decisions made by the multidisciplinary child protection team regarding substantiation of abuse. This included using an ecological framework to identify factors at the level of the individual child, the family, neighbourhood, ethnicity and society that might intersect and contribute to recurrence of abuse; for example, by extracting data on child and carer vulnerabilities, and using anonymised postcode data to establish whether geographic location and socio-economic status are risk factors [16,17]. 

## 2. Materials and Methods

### 2.1. Study Design

A retrospective chart review of all children aged 0–16 years referred to Birmingham Children’s Hospital with suspected child abuse or neglect between July 2011–September 2014 was undertaken. Data were extracted from the Child Protection referral form, information held by the hospital’s Child Protection Team and the child’s hospital record; anonymised using a unique research ID number; transferred to an Excel spreadsheet; and retained in a password protected, secure file, in accordance with the Data Protection Act (1998) and the General Data Protection Regulation (GDPR, 2018). The list of cases and their ID number was held by a member of the Child Protection team in a password protected file. No identifiable personal data were used; researchers handling extracted data were unable to identify any of the cases by name. The NHS Health Research Authority approved this study (Proportionate Review Sub-Committee of the North West—Greater Manchester South REC, reference 16/NW/0679).

### 2.2. Data Collection

The database paralleled the child’s journey from presentation or transfer to the hospital to discharge in order to reduce input error and allow easier centralisation and access to data [18]. Data extracted included date of incident, demographics (age, gender, number of siblings and their age, ethnicity, home postcode, disability), the child’s main carer, first language and ethnicity of carer, carer vulnerabilities such as domestic violence, alcohol and substance misuse, mental health concerns and learning difficulties, description of the child’s presenting injuries or concerns, previous involvement with Children’s Social Services, child’s assigned child maltreatment category at initial multiagency case conference, hospital length of stay, discharge location and follow-up with Children’s Social Care. 

The data were abstracted by members of the research team, all with expertise in sensitive/child research based at Birmingham Children’s Hospital or the University of Birmingham. There was prior agreement on how each variable would be captured, where it was located in the referral forms and child’s record, and how it would be recorded in order to establish explicit coding criteria to increase inter-rater reliability of data abstraction [19,20]. Random checks were undertaken by one member of the team to ensure reliability in the process [18]. Meetings were held regularly to resolve any ambiguous or conflicting data.

### 2.3. Measures

Substantiated case decision: A referral of a child with possible abuse and/or neglect was deemed to be substantiated if a multiagency child protection conference judged that there were reasonable grounds to suspect that the child may continue to suffer, or is likely to suffer, ‘significant harm’, i.e., ill treatment or impairment of health and development by a carer as compared with the care that would reasonably be expected of a similar child [3].

Co-occurring childhood adversities within the family domain: Data on whether the child known to statutory bodies, particularly Children’s Social Services, prior to, or at the time of presentation, and documented domestic violence, parental drug and alcohol misuse and mental health concerns or learning disability in one or both parents or carers was collected, as these are known to be associated with child maltreatment.

Typology of neglect: Neglect is the omission of caretaking behaviour that a child needs for healthy development [21]. Where neglect was identified as the primary presentation, or was identified as being a major contributing factor to the presentation and referral, one of the authors (G.D.) assigned this to one of the typologies described by Knutson et al. [22] and modified by Mennen et al. [21], based on the information available in the child’s record:-Care (or physical) neglect: carer fails to provide for the child’s basic needs, such as food or clothing;-Environmental neglect: when a serious health and safety hazard is present in a child’s physical surroundings or the house is not adequate in size or cleanliness;-Medical neglect: failure to provide appropriate medical care when a child is in need of a medical assessment or treatment for injury, illness or disability;-Educational neglect: carers failing to send the child to school or prevent the child from having a suitable education;-Supervisory neglect: concerns a situation where a carer leaves a child alone or in inappropriate substitute care.

Some authors, such as Welch et al. [23], place emotional abuse—when children are deprived of their emotional needs (forming secure, positive attachments with adults)—within the topography of neglect. 

Categories of abuse for children placed on a Child Protection Plan are those defined in Working Together to Safeguarding Children [24]: physical abuse, neglect, emotional abuse, including witnessing domestic violence and sexual abuse. The data on substantiated cases did not capture all children on a Child Protection Plan. Where possible, categories were assigned by one of the authors, based on presenting characteristics. This did not account for any children with multiple categories of abuse.

IMD deciles: The Index of Multiple Deprivation (IMD) is the UK Government’s official measure of relative deprivation for neighbourhoods in England, and covers the domains of income, employment, education, health, crime, barriers to housing and services and the living environment [25]. It ranks every small area of around 650 households (Lower Super Output Area) in England from 1 (most deprived) to 32,844 (least deprived). The English IMD Postcode checker was used to determine the Deprivation decile for the postcode of each case where available [26], with Decile 1 being the most deprived 10% of neighbourhoods and Decile 10 the least deprived 10%. 

Ethnicity: The list of ethnic groups was taken from that used in the Office of National Statistics’ (ONS) 2011 census. One’s ethnic group is self-defined and there is a ‘prefer not to say’ option when asked. The figures for ethnicity in Birmingham were taken from respondents to the 2011 census.

In this paper, ‘Carer’ is used to include any primary caregiver.

### 2.4. Data Analysis

Demographic and injury data have been categorised and reported as a number (percentage). Pearson’s χ^2^ test was used to test the distribution and significance of these variables. *P* values are derived from Pearson’s χ^2^ test. A two-sided statistical significance level of 0.05 was applied to all results. Data were nominal, apart from age, which was recorded as a ratio. Ratio data were transformed to original data to aid with statistical analysis. All analyses were conducted with SPSS (version 25.0, IBM Corp, Armonk, NY, USA).

## 3. Results

There were 531 cases identified during the study period (July 2011—end September 2014), of which 452 were available for analysis, 83 cases being excluded due to insufficient data or not involving the Child Protection Team. This represents around 1% of all unplanned admissions to the hospital during the study period, with an annual overall incidence of 0.3/1000 children from within the Birmingham area. In total, 162 cases (35%) were from outside the Birmingham area, 114 of which were transferred from other acute paediatric units to Birmingham Children’s Hospital for further investigation and management. The majority of cases were from the West Midlands region, with 49 (11%) from outside.

There were 246 (54%) males and 206 (45.6%) females in the sample; 146 (32%) were less than 1 year of age, 202 (45%) between 1 and 5 years of age, 33 (7%) between 5 and 11 years and 71 (16%) between 11 and 16 years of age, respectively. Child maltreatment was substantiated in 294 (65%) referred cases and not substantiated in other 158 (35%). 

The results are presented aligned with the ecological framework, from child through to family through to neighbourhood and wider society. 

### 3.1. Geographic Mapping and Socioeconomic Profile

The majority of cases of both substantiated and unsubstantiated cases were from the most deprived 10% of neighbourhoods in England, with a sharp downward gradient to the least deprived neighbourhoods (Figure 1).

We used the postcodes to create a geographic heat map in Microsoft Excel (Figure 2). Although there was a scatter of cases throughout Birmingham and neighbouring localities, the largest number of cases in the hospital’s immediate ‘catchment area’ were from Area A (20 cases) and Area B (23 cases) (Figure 3). These areas have contrasting topographies: Area A, in the inner city, is characterised by widespread socioeconomic deprivation, with the vast majority of families residing in 1st decile level neighbourhoods, where 19 of the 20 cases were from. Area B, however, on the southern city boundary, has small pockets of severe deprivation, in amongst larger areas of lesser deprivation, ranging from 2nd–6th decile. The majority of cases (18 of 23) lived in 1st decile neighbourhoods.

### 3.2. Ethnicity

The ethnic distribution for substantiated and unsubstantiated cases, in comparison to 2011 census data, is shown in Table 1. Ethnicity was either not stated or not documented for 43 cases (9.5%). Thus, the figures shown are based on 409 cases. There was an over-representation of Eastern European, Black-Caribbean and Black-African children, and an under-representation of Asian-British backgrounds, but this did not reach significance for case substantiation.

In 351 (77.7%) of referred children, the first language of parents/carers was English, 21 (4.6%) Urdu and 9 (2.0%) Polish, compared with 84.7%, 2.8% and 0.9% in the 2011 Census, respectively. Overall, 23 languages other than English were represented.

### 3.3. Potential Predisposing Factors to Child Abuse and Neglect within the Family Domain

#### 3.3.1. Parental Vulnerability

In the vast majority of referrals (96%), co-occurring parental ‘vulnerabilities’ were recorded: Domestic violence occurred in 104 (24%), drug and alcohol misuse in 57 (13%) and mental health concerns in 63 (14%) of 433 cases. Domestic violence alone occurred in 52 (12%). In total, 28 (6%) had two, and 8 (1.8%) three parental vulnerabilities. Five parents had documented learning difficulties.

#### 3.3.2. Previous Involvement with Children’s Social Care

There had been previous involvement with statutory services, predominantly Social Services, in 211 (46%) of referrals, 28 of whom were on a Child Protection Plan at the time of admission; (this might be an under-estimate, as there were missing data in 58 cases). Four children were in care at the time of presentation and parents were care leavers in one.

#### 3.3.3. Childhood Disability or Chronic Health Condition

Seventy-eight (17%) of children referred had a documented physical disability (*n* = 4), learning disability (*n* = 18), emotional or behavioural difficulty (*n* = 16), chronic health condition (*n* = 20) and not stated (*n* = 20).

Data on the frequency of conversion to substantiated abuse for disability or chronic condition are shown in Table 2. This was highly significant where domestic violence and previous involvement with statutory bodies were a feature. Gender was not a significant contributor.

### 3.4. Presenting Concerns

The reason for referral for each child to the Child Protection Team is shown in Table 3 and Figure 4. Overall, physical injury accounted for 270 cases (60%). The high frequency of burns (21%) in relation to bruising (15%), head injury (10%) and extremity fractures (8%) reflects the hospital’s designation as a major tertiary Burns Unit; 51 cases (55%) were transferred from other hospitals for opinion and management.

Soft tissue injuries (39%) were predominantly bruising (9.8%), lacerations (5.4%) and there were 15 (3.3%) cases of dog bites. Fractures were predominantly upper and lower extremity fractures, with only one case of occult rib fracture. However, skeletal survey was positive for occult skeletal injury in nine cases. Head injury cases were divided between scalp haematoma, with or without underlying skull fracture (*n* = 19) and intracranial haemorrhage (*n* = 15), predominantly subdural haemorrhage (*n* = 15). In total, 18 of the 44 cases of head injury were transferred in from another hospital for neurosurgical review.

The 31 children with ‘other’ injuries were mainly multiple traumas from falling from upstairs windows (*n* = 8); motor vehicle accidents, car or pedestrian (*n* = 13), where supervisory neglect was a feature; stairway injuries (*n* = 3); and various other presentations, such as assault in a public space, and degloving and other injuries from being exposed to dangerous situations.

### 3.5. Neglect and Emotional Abuse

Overall, there were 157 (35%) cases of neglect and 68 (15%) with emotional abuse. The 56 (12%) cases of primary neglect represented in Table 3 were concerns triggered by admission (e.g., malnutrition due to care neglect) or arose during an admission for an intercurrent illness or for elective surgery; 41 were medical neglect, 36 of which were substantiated. Medical neglect was associated with failing to secure medical attention for an injury, not bringing a child for medical appointments, refusing medical treatment for the child and inappropriate delay in seeking medical attention for an injury. There was one case of ‘spirit possession’. There were 10 situations involving environmental neglect and 3 care neglect, all of which were substantiated.

In total, 24 of the 36 cases of emotional abuse as the main presenting concern, related to current exposure to domestic violence (witnessed on ward by child or staff, disclosed or child involved but no injury found); six were admitted for a ‘place-of-safety, including one survivor of forced marriage and two of child sexual exploitation.

Neglect and emotional abuse were judged to be a contributory, but not the primary factor in 101 (23%) and 31 (7%) of referrals, respectively. These were mainly burns and self-harm cases (Table 4); there were no cases in the bruising, head injury or skeletal injuries groups. In total, 14 of 16 children with dog bite marks we considered to be due to supervisory neglect.

From a total of 93 presentations with burns, 63 (68%) were adjudged associated with neglect, 57 of which were supervisory neglect, including 2 housefire incidents, with 40 cases substantiated. The breakdown for each type of burn is shown in Table 4. There were no cases of educational neglect.

There were 35 admissions with self-harm, predominantly adolescents. In total, 30 (86%) were associated with emotional abuse, substantiated in 10. Historical sexual abuse or sexual exploitation occurred in 8 substantiated cases and 3 unsubstantiated cases.

### 3.6. Categories of Abuse

The abuse categories for substantiated cases in this study (*n* = 294), compared with data for England and Wales for those children on a Child Protection Plan [12], are shown in Table 5. Not all cases could be assigned a category due to lack of data. Within the category of emotional abuse, there were 24 cases of children witnessing domestic violence.

## 4. Discussion

The 452 children admitted with suspected abuse or neglect represents an annual incidence of 0.3/1000 unplanned admissions for all age groups during the 3-year study period (2011–2014), a rate similar to that found in England in 2009 by Gonzalez-Izquierdo et al. [7]. This represents 0.98% of all unplanned admissions to the hospital during the study period, and 0.2% in infants < 1 year of age. Differences in populations studied, including age range and typology of injuries, methodology and jurisdictions notwithstanding, these data are broadly comparable with other high-income countries such as France (0.1% for infants < 1 year admitted to all hospitals from 2007–2014), the Netherlands (0.2% in children < 18 years) and the USA (0.17% in children < 24-months) [27,28,29].

While these data measure the frequency of referred cases to hospitals, many more children with suspected maltreatment are assessed within community settings. Moreover, in their meta-analyses of child maltreatment cases reported to agencies globally, Stoltenborgh et al. [30] found that the prevalence rates for referred cases, similar to those above, were much lower than from studies of self-reported maltreatment. Thus, intercountry or interjurisdictional comparisons would be greatly enhanced if there was an agreed operational definition and minimum dataset for child maltreatment across both hospital and community settings, combined with longitudinal data linkage studies.

### 4.1. Case Substantiation

The conversion rate from referred, as yet unsubstantiated cases to cases substantiated for abuse or neglect could also be used as an inter-country or interjurisdictional benchmark. A case is substantiated if investigations find sufficient credible evidence of harm or maltreatment. In this study, child maltreatment was substantiated in 65% of referred cases, very similar to single centre, tertiary children’s hospital data from Zurich (61%) and Ankara (67%) [31,32], but less than for whole population data for all types of abuse from the UK (47%) and Ontario, Canada [44%] [12,33]. This difference might well be due to the higher likelihood of significant injury in children referred to hospital centres.

Investigations are generally undertaken by multidisciplinary teams, as quoted in this study and other studies. The substantiation rate, therefore, could be regarded as a proxy measure for the accuracy of risk assessment and decision-making capabilities of multidisciplinary teams. This was beyond the scope of this study. Using a Delphi approach, Kistin et al. [14] found that multidisciplinary teams are effective when they share a systems approach that provides important scaffolding to decision-making, have team ‘collegiality’ and cohesion, and when they are well-resourced and well-staffed. Using these data, they developed and tested a novel tool for team self-evaluation [34].

However, some authors have questioned whether substantiation is a reliable or valid operational metric for child abuse, particularly for research, policy and practice; in one study, medium-term outcomes for substantiated cases in respect of a child’s behavioural and developmental status and number of new child maltreatment reports were statistically indistinguishable from those for unsubstantiated cases, and that any distinction might be the result of reporting bias or poor decision making [35]. They argue that all referrals to child welfare agencies have a service need, the provision of which would result in better outcomes. This is in keeping with a model of child protection that replaces risk with care as an organising typology of child protection [36], a typology used in many jurisdictions [2].

Case characteristics associated with case substantiation include ethnicity, socio-economic factors (such as neighbourhood deprivation and family income), carer health and functioning (such as mental health concerns, drug and alcohol misuse and domestic violence and abuse), the family’s previous involvement with child protection agencies (including a carer’s history of abuse), a single carer, ‘re-ordered families’, poor social networks and type and severity of injury [13,33,37]. Notwithstanding the above critique, our study was designed to capture data on these characteristics.

### 4.2. Ethnicity

In this study, the ethnic categories were those used in UK Census data and are based on self-identification by respondents. Overall, there was an over-representation of Eastern European, Black-Caribbean and Black-African children, and an under-representation of Asian-British backgrounds when compared with the 2011 Census [15]. This did not reach significance for substantiated cases. There are no single-centre studies in the UK with which to compare these data and it is difficult to find inter-country comparisons, as there is so much variation in composition and characteristics of ethnic groups.

However, researchers from the USA and Canadian have found over-reporting of African-American and native American or First Nations children, and under-reporting of Hispanic and Asian-American children for child maltreatment in comparison to White children, which is not always reflected in substantiation rates. These studies reveal a complex and little-understood relationship between child maltreatment and ethnicity, individual and neighbourhood poverty and population density and racial bias [38,39,40,41,42].

### 4.3. Neighbourhood Poverty

In this study, the majority of reported and substantiated cases were from the most deprived neighbourhoods in the UK. Heat maps revealed the largest numbers residing in deprived, high-density inner-city neighbourhoods, with smaller numbers in pockets of lower density, equally deprived lower density neighbourhoods within a larger, lesser density locality. The data were not able to demonstrate any coincidence between ethnic groups and these deprived neighbourhoods, but Cangiano [43] was able to do so in mapping ethnicity and poverty in Birmingham.

Researchers, predominantly from the USA, have been using an intersectoral approach and ecological framework to unravel the complexities of how neighbourhoods shape social processes that contribute to child maltreatment and community violence [17,39]. The pioneering ‘social ecology’ studies by Garbarino and Sherman [44], Garbarino and Kostelny [45] and Sampson et al. [46] found that, compared with ‘low-risk’ neighbourhoods, there was a sense of dissociation from the community, less reciprocal exchange and lack of social cohesion in ‘high-risk’ neighbourhoods, where child maltreatment rates were linked with chronic unemployment, overcrowding, residential instability, visible criminality, physical desolation and lack of informal social support networks. These and other studies have been well-reviewed by Coulton et al. [47]. Studies have also demonstrated the potential protective effect of developing neighbourhood efficiencies and social capital on violence and child maltreatment [48].

### 4.4. Potential Predisposing Factors to Child Abuse and Neglect within the Family Domain

There is strong evidence for an association between maternal domestic violence and abuse, parental drug and alcohol misuse and maternal mental health and child maltreatment [49,50,51]. The pathways to child maltreatment are complex and these associations might reflect a broader context of exposure and outcome, with many potential confounders. Drawing on models of probabilistic causation, Munro et al. [52] demonstrated that such risk factors are neither necessary nor sufficient conditions for child maltreatment to occur. Yet, there is accumulating evidence of intergenerational continuities in child maltreatment, mediated by unresolved trauma, social isolation and poor parenting styles, and expressed as dissociation or post-traumatic stress disorder, particularly in the mother, and by domestic violence [53,54,55,56].

In this study, a history of domestic violence and abuse existed in 24%, parental mental health concerns in 14% and parental drug and alcohol misuse in 13%. Only domestic violence and abuse reached significance for substantiated cases. This finding accords with studies of the adverse effects on children exposed to family domestic violence throughout their life course [57]. Using data from a longitudinal birth cohort study in Queensland, Ahmadabadi et al. [49] found the association between maternal domestic violence and abuse and maltreatment of offspring was independent of a wide range of ecological and familial confounders except for father’s history of mental health problems. Our data did not distinguish between maternal and paternal risk factors. While the study data included the index child’s previous involvement with statutory bodies, a history of carer maltreatment as child was not. This is a major omission, as it is highly predictive of their offspring’s maltreatment [53,56].

### 4.5. Child’s Previous Involvement with Statutory Bodies

There was previous involvement with statutory services in 48% of index children and this was a significant factor for case substantiation. This is a higher frequency than found in a systematic review [58], where recurrence rates were in the region of 20% over a 2-year period. Recurrence risk was associated with neglect, multiple episodes of maltreatment, children with special educational needs, carer alcohol and drug misuse and mental health concerns, domestic violence, lack of social support, lone parenthood and poverty. Bae et al. [59] found that maltreatment risk recurrence was also associated with lack of contact with children’s social care and other aspects of service provision. The higher frequency of previous statutory involvement in this study might reflect sample bias.

### 4.6. Presenting Concerns

Physical injuries (60%) were the main reason for referral, with burns (21%) and soft tissue injuries (15%), mainly bruising (9.8%), followed by head injury, extremity fractures and major trauma. This is double the overall total of 31% in another comparable tertiary centre over a similar period [31]. This is due mainly to the predominance of burns, and thus, reflects sample bias, as Birmingham Children’s Hospital is a major regional tertiary Burns Unit.

The relatively high number of presentations with dog bites prompted the hospital to develop a specific child protection pathway for their management. Presentations of children with domestic dog bites has risen by 17% in the UK from the end of our study period to 2018. Jakeman et al. [60] drew attention to the severe nature of the physical injuries (predominantly facial) and the long-standing psychological harm to children from dog bites.

There were only 3.5% cases of suspected child sexual abuse. This is very low in comparison to other single centre studies—e.g., 38% in Zurich [31]—and reflects sample bias, as such cases were assessed by a multiagency team child sexual assault referral unit.

### 4.7. Neglect

Overall, neglect occurred in 35%. This is a similar frequency to England and Wales but greater than that for other single centre studies such as Zurich and Ankara (19% and 12%, respectively) [31,32]. In 56 cases (12%) where neglect was the primary presentation, the vast majority were due to substantiated medical and environmental neglect. Parmeter et al. [61] found 61 cases (4%) of medical neglect in admissions to a single tertiary children’s centre in Sydney. These cases involved children with chronic, complex medical conditions, where carers were deemed unwilling to follow medical advice or unable to provide the necessary care, situations similar to our cases.

In addition, there were 101 (25%) of cases where neglect was linked to the presenting condition, in particular, to burns, falls from windows and motor vehicle injuries. The vast majority of spill scalds, contact burns and flame burns (40 cases) were associated with substantiated supervisory neglect, with 3 being due to medical neglect (inappropriate delay in presentation). Although the total number of burns admissions to Birmingham Children’s Hospital during the study period was not known, Chester et al. [62] found that neglect was associated with 9% of burns admitted to the same hospital in 2006. James-Ellison et al. [63] found that children under 3 years of age with burns of any category are at risk for abuse or neglect before the age of 6 years, emphasising the need to carefully assess all young children with burns, as having significant on-going service needs.

### 4.8. Emotional Abuse

Emotional abuse was found in 17%, lower than for England and Wales (33%) and Zurich (38%) [31]. This was the primary concern in 36 cases, exposure to domestic violence at the time of presentation being the predominant reason. There were 31 cases where emotional abuse was associated with deliberate self-harm, and, in 8 cases, this was linked with historic child sexual abuse and sexual exploitation, with one case of forced marriage. The emotional burden due to, at times, undisclosed sexual abuse and exploitation is considerable. The pathways from trauma to self-harm are complex but are mediated though mechanisms such as post-traumatic stress/dissociation and emotional dysregulation [64].

### 4.9. Limitations and Strengths

This study was an attempt to move away from measuring the extent of child maltreatment using a unitary coding system for a single episode, with its tendency to underestimate the number of cases, towards data capture to represent a child’s trajectory from referral to initial outcome. This has provided a rich source of data from predisposing risks, particularly domestic violence and the child’s prior contact with statutory agencies to various typologies of neglect involved in presentations such as burns. Geographical mapping illustrated the extent of poverty in referred cases.

However, a single-centre study does have major drawbacks. Retrospective data capture from an individual child’s record is fraught with missing data. In this study, large numbers of cases could not be analysed due to missing data; there were no data on a carer’s childhood experience of maltreatment, a significant predictor of child maltreatment. Sample bias was another limitation. For example, the hospital is a superregional children’s burns unit, and this gave a preponderance of burns admissions. The hospital does not assess children with suspected historic or acute sexual abuse (CSA), as they are referred to a regional Paediatric Sexual Assault Referral Centre. The hospital is a tertiary centre. This might account for the relatively high numbers of children transferred from other hospitals, with more severe injury. This makes inter-hospital benchmarking problematic. In addition, only a minority of cases of suspected child maltreatment require hospital admission, providing further sample bias.

Although the study demonstrated a link between neighbourhood deprivation and child maltreatment, it could not investigate the intersectoral and other dimensions in the complex interrelationships between poverty and child maltreatment [65].

## 5. Conclusions

This and other studies attest to the need for an agreed dataset to discover how best to keep children safe, across jurisdictions and countries with differing child protection typologies, in order to learn from each other. This study shows how data extracted from case notes can provide a view of a child’s trajectory from referral to immediate safety, but that is, by its very nature, a restricted and short-term view. There is an urgent need to agree a minimum dataset and codes, linked longitudinally to social services and police data to measure a child’s trajectory through the child protection system and beyond, to permit cross-jurisdictional and inter-country comparisons and to develop accurate prediction modelling techniques to accurately estimate risk and safety, and to plan and evaluate interventions [66,67].

## Figures and Tables

**Figure 1 ijerph-19-08213-f001:**
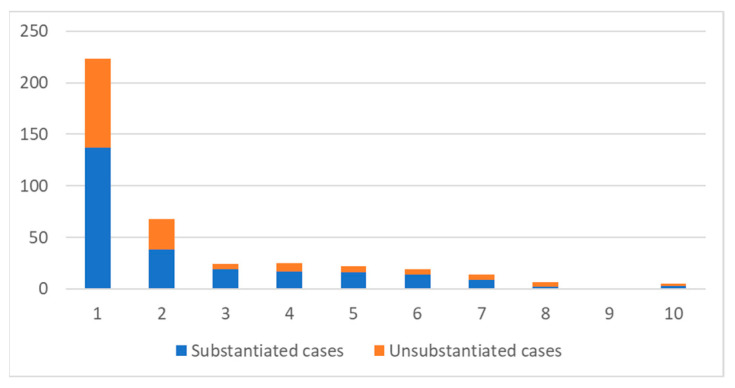
Number of cases in each IMD decile for neighbourhood postcode.

**Figure 2 ijerph-19-08213-f002:**
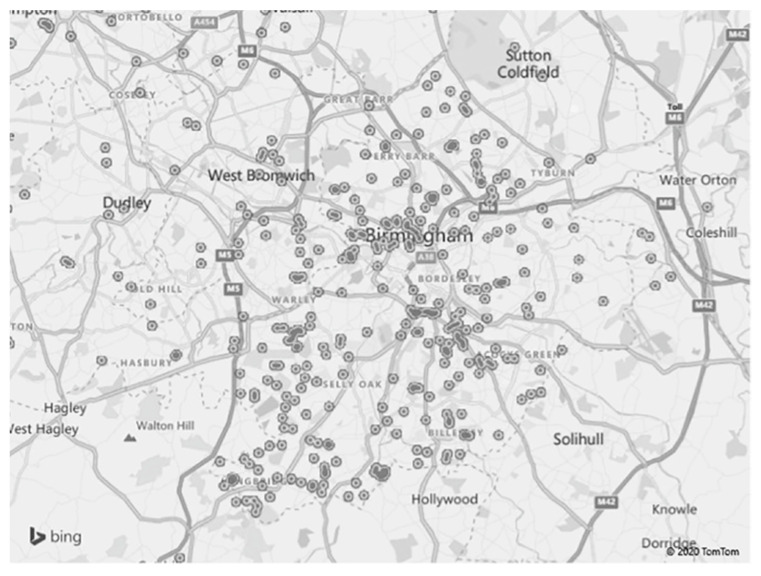
Heat map showing distribution of cases in Birmingham and surrounding areas.

**Figure 3 ijerph-19-08213-f003:**
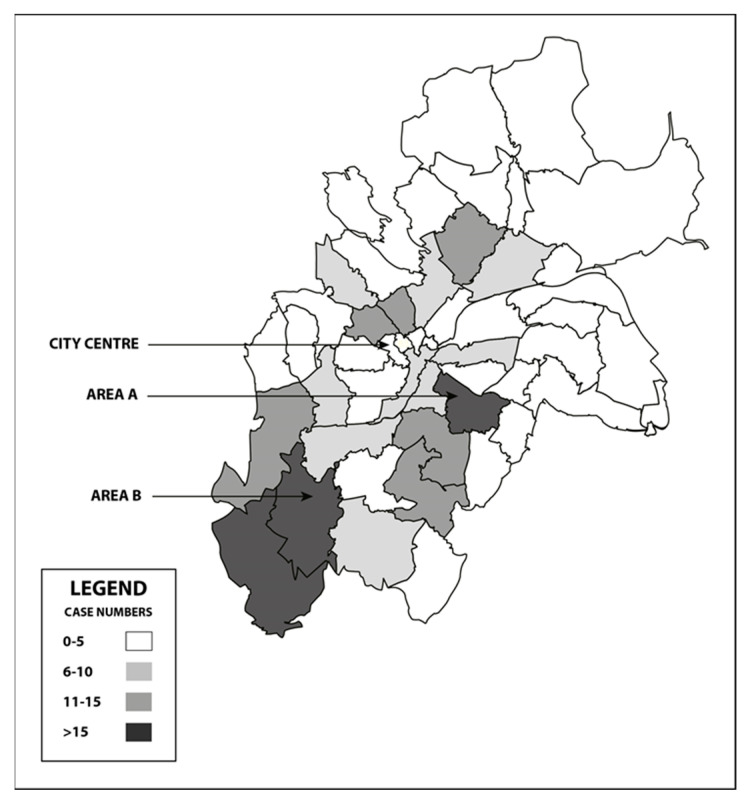
Distribution of cases within Birmingham Postcode areas.

**Figure 4 ijerph-19-08213-f004:**
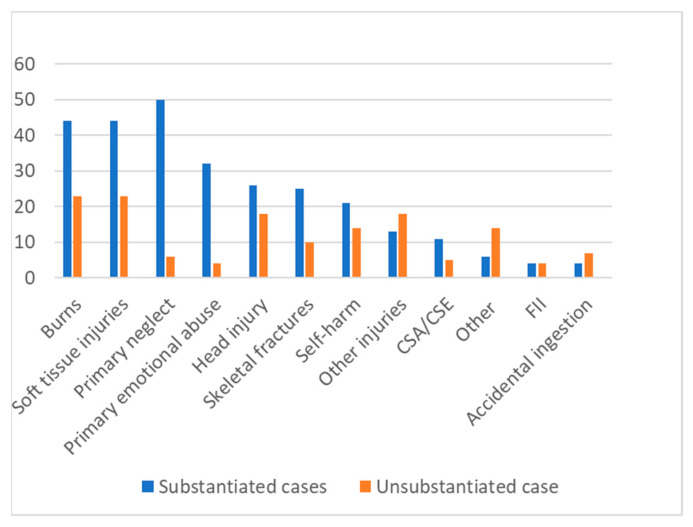
Referral characteristics of the sample.

**Table 1 ijerph-19-08213-t001:** Ethnic distribution of cases compared with 2011 census data.

Ethnic Category	Substantiated Cases		Total No. of Cases		2011 Census Birmingham
	Yes	No	No.	%	%
White					
- British	127	90	217	53.0	53
- Irish	1	0	1	0.2	2
- Eastern European/Any other white background	10	11	21	5.1	2.7
Mixed or multiple ethnic groups					
- White and Black Caribbean	9	2	11	2.7	2.3
- White and Black African	2	0	2	0.5	0.3
- White and Asian	1	3	4	1.0	1.4
- Any other mixed background	5	2	7	1.8	0.8
Asian or Asian British					
- Indian	6	3	9	2.3	6.0
- Pakistani	31	13	44	10.8	13.5
- Bangladeshi	7	3	10	2.4	3.0
- Any other Asian background	9	8	17	4.2	0.8
Black and Black British					
- Caribbean	19	11	30	7.3	4.4
- African	12	7	19	4.6	2.8
- Any other Black British background	2	2	4	0.9	1.8
Any other Ethnic Group	7	6	13	3.2	1.0
Total	248	161	409	100	100

**Table 2 ijerph-19-08213-t002:** Potential predisposing factors to abuse in referred cases.

Potential Predisposing Factor	Substantiated Cases		Total No. of Cases		*p*-Value
	Yes	No	No.	%	
Parent/carer ‘vulnerability’					
- Domestic violence	76	28	104	24	0.004
- Drug and/or alcohol misuse	36	21	57	13	n.s.
- Mental health concerns	41	22	63	14	n.s.
Childhood disability or chronic illness	44	35	79	17	n.s.
Previous involvement with statutory body	140	71	211	48	0.001

n.s. = non-significant.

**Table 3 ijerph-19-08213-t003:** Referral characteristics of the sample.

Reason for Referral	Substantiated Cases		Total No. of Cases	
	Yes	No	No.	%
Soft tissue injuries (e.g., bruises, lacerations, bites)	44	23	67	14.8
Burns	58	35	93	20.5
Skeletal fractures	25	10	35	7.7
Head injury	26	18	44	9.7
Other injuries (e.g., multiple trauma, falls from heights, stairway falls)	13	18	31	6.8
Self-harm	21	14	35	7.8
Accidental ingestion	4	7	11	2.4
Fabricated or induced illness	4	4	8	1.8
Primary Neglect	50	6	56	12.4
Primary Emotional abuse	32	4	36	8.0
Sexual abuse/exploitation	11	5	16	3.5
Other (e.g., BRUE, oronasal haemorrhage)	6	14	20	4.5
Total	294	158	452	100

**Table 4 ijerph-19-08213-t004:** Cases of neglect and emotional abuse associated with injury presentation.

	Supervisory Neglect		Care Neglect		Medical Neglect		Emotional Abuse		Environmental Neglect		Total Neglect
	S	NS	S	NS	S	NS	S	NS	S	NS	
Dog bites	5	8							1		14
Burns (Total)											
- Scald	14	5			1					1	21
- Immersion	3	1			2					1	7
- Contact	11	4									15
- Chemical	2										2
- Flame	6	3							1		10
- Other	4	4									8
Falls from windows	3	5									8
Motor vehicle injury	6	1						1			8
Self-harm							21	9			30
Ingestion	4	4				1					9
	58	35	0	0	3	1	21	10	2	2	132

S = Substantiated; NS = Not substantiated.

**Table 5 ijerph-19-08213-t005:** Categories of abuse (*n* = 401).

Category of Abuse	Study		England & Wales (%)
	No.	%	
Physical	166	41.4	9.8
Neglect	157	39.1	41.9
Emotional abuse (including witnessing domestic violence)	67	16.7	32.8
Sexual abuse	11	2.7	4.5
Multiple categories	Not known		11.0
Total	401		100

## Data Availability

Data were obtained from Birmingham Children’s Hospital and are available from Geoff Debelle g.debelle@nhs.net with the permission of Birmingham Children’s Hospital.

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
