# Peer review of "The Typology and Topography of Child Abuse and Neglect: The Experience of a Tertiary Children’s Centre"

_ijerph, 2022, doi:10.3390/ijerph19138213_

Round 1

Reviewer 1 Report

Abstract

“Good child protection systems…” – Perhaps use concise language “Effective child protection systems…”

Line 17 – 20: It sounds like implications of the study are included in the study description.

Single centre data system is identified at the end of the abstract, but not included earlier, which results in some confusion.

Overall, the abstract should be edited to be clearer.

Introduction

Line 76: What is “child projection plan”? is this supposed to be “child protection plan”?

Line 76-77: This reference needs to be identified further (in the references section). This statistic seems very high, I therefore attempted to validate via the reference, it is unclear however, where this reference can be located and might be from the education system.

Line 87: This line and onward seems like it might be a background or literature review. If so, there should be a header to indicate.   

Materials and Methods

Line 124 (and onward): Data is plural.

Measures

Line 180: Was emotional neglected included in this study?

Overall, this section was confusing. Were all measures dichotomous? Some measures were described in more details than others.

Results

Nothing in the data analysis section indicates how the map (figure 2) was developed.

The results section was very confusing and could have used more structure.

Discussion

Line 464: This paragraph suggests that overrepresentation has been identified in US and Canada, yet there are no Canadian studies included in the citations (38-42).

Line 493: This line might be a mistake.

Overall, this study’s link between research question - methodology – knowledge claim was very confusing. It is stated that case data was used to assess decisions by a multidisciplinary CP team. It seems as though, however, only the substantiation decision was examined. Further, an ecological perspective was used as a foundation for the research design. This is a logical examination of data; the research design however, did not address an ecological systems approach. Chi-square provides relational results and only implies association – on a single level. Ideally, multilevel modeling would be used to answer the research question with an ecological framework. Lastly, while SES was indicated in an aim of the study, it was sparsely discussed in the results/discussion.

Reviewer 2 Report

This is a very interesting paper on the types and topography of child abuse and neglect in a tertiary children's centre. 

The paper is focused on the experience in this center and opens new avenues on the research and public health policies on child protection. The paper is well-written and is of interest for the readers. However, prior to its publication, I recommend several changes.

The third paragraph of the paper emphasize that the World Health Organization and the International Society for Prevention of Child ABuse and Neglect call for an operational definition of child maltreatment. In this paragraph, I recommend to clarify that the definition should be recommend despite the occurrence of mental health symptoms by itself.

The main aim of the study is to determine what data could be extracted from hospital records and which factors should be identified from the child, family and neighbourhood, ethnicity, etc. I would also recommend to add, as an objective, to propose health public policies as a main objective after discussion of results. This could be also a third objective.

The authors reported that 65% of referred cases were substantiated, and 35% not substantiated. Was there any follow up to confirm the 35% of cases? 

The frequency of sexual abuse is 2.7% in patients with abuse. Is this percentage comparable with the current scientific literature? In some cases, different types of neglect or abuse appear as combined. 

Emotional abuse was found to be 17%, and to be Lower than for Entgland and Wales, and Zurich. How is the comparison with studies done in the United States?

Soft tissues injuries like bruises, lacerations and bites are reported in a same group. Would it be possible to have the number of patients separately?

The last paragraph of the discussion section is called "Strenghts and weaknesses". I prefer to rename if as "Limitations and Strenghts".
